# Movement Pandemic Adaptability: Health Inequity and Advocacy among Latinx Immigrant and Indigenous Peoples

**DOI:** 10.3390/ijerph19158981

**Published:** 2022-07-23

**Authors:** Mario Alberto Viveros Espinoza-Kulick

**Affiliations:** Ethnic Studies, Cuesta College, San Luis Obispo, CA 93405, USA; mario_espinozakulick@cuesta.edu; Tel.: +1-(805)-904-9225

**Keywords:** COVID-19, health equity, Latinx health, Latinx indigeneity, language, health access

## Abstract

The COVID-19 pandemic exacerbated longstanding inequities in resources and healthcare, stacked on top of historical systems that exploit immigrants and communities of color. The range of relief, mutual aid, and advocacy responses to the pandemic highlights the role of social movement organizations in addressing the ways that immigration status creates systemic barriers to adequate health and wellbeing. This paper conceptualizes what I call, “movement pandemic adaptability,” drawing from a decolonial-inspired study including participant-observation (September 2018–September 2020), interviews (*n* = 31), and focus groups (*n* = 12) with community members and health advocates. Data collection began before the COVID-19 pandemic (September 2018–February 2019) and continued during its emergence and the initial shelter-in-place orders (March 2019–September 2020). Movement pandemic adaptability emerged as a strategy of drawing from pre-existing networks and solidarities to provide culturally relevant resources for resilience that addressed vulnerabilities created by restrictions against undocumented people and language barriers for communities that speak Spanish and a range of Indigenous languages. This paper presents how the relationship between immigration status and health is influenced by the local context, as well as the decisions of advocates, policymakers, and community members.

## 1. Introduction

Groups that advocate for immigrants and immigrant rights rely on creative direct action strategies to organize for change [1,2]. Similarly, collective efforts related to addressing health inequities often utilize social movement strategies to target healthcare providers, insurance companies, and other aspects of the medical industry directly [3,4,5,6,7,8]. Scholars have documented major inequities in healthcare access and health outcomes within immigrant communities [9,10,11,12]. However, there has been little research examining the connections between immigrant health and advocacy movements. This paper bridges these research areas to analyze how immigrant health advocacy responded to the shifting political, social, and environmental context brought about by the COVID-19 pandemic.

Like other large-scale social and environmental events, the onset of a global pandemic creates a cascade of social changes [13,14,15,16,17,18]. The emergency response creates openings within the broader political setting for groups to amplify efforts for change [19,20]. Advocates draw on existing networks and communities of support to mobilize effectively during times of renewed opportunity [21]. Immigrant health movements, in particular, are accustomed to operating in limited-resource environments with political hostilities, leading to a range of creative and efficacious strategies [15]. However, researchers have yet to identify a conceptual framework for analyzing how immigrant health movements can inform broader work for health equity and access to care.

Unequal access and discrimination have systematic effects on the wellbeing of communities most at risk for disease [22]. Vocal xenophobic political ideologies and anti-immigrant policies—such as family separation, the widespread use of militarized enforcement raids, deportation, and detention—contribute to a culture of fear and reproduce health disparities [16,23,24]. These inequities have widespread effects on immigrant and Latinx communities, including mental health and chronic disease at the individual-, family-, and community-level. Academic researchers have spent considerable resources to document these inequities. With this knowledge well in hand, scholars can move beyond observing racial/ethnic disparities to document tenable strategies for ending systemic racism and its adverse effects on healthcare systems and practices [25,26,27]. This paper contributes to those efforts by demonstrating the interlinkages between immigrant advocacy and health advocacy [1,2,4,5,7,8,20,24,28,29,30,31,32,33,34,35].

Immigrant advocates have long contended with the risks facing immigrants [23,24]. Deportation, detention, and other punitive policies make direct action inherently risky for undocumented individuals and mixed-status families. However, “undocumented and unafraid” young people have also leveraged that risk to effectively gain visibility for policy change [30,31,36,37,38]. These creative responses to high-risk environments have prepared immigrant health advocates to lead equitably during the pandemic by prioritizing those most affected and at risk.

The concept of “movement pandemic adaptability” developed in this paper highlights how groups and organizations can lead through and beyond the COVID-19 pandemic with a central focus on equity. This theory emerged from a decolonial-inspired study of Latinx immigrant health advocacy on California’s Central Coast. Movement pandemic adaptability includes building culturally responsive and language-accessible resources for community support, understanding opportunities in a changing world, identifying policies that impact communities at multiple levels, and drawing from localized expertise.

## 2. Materials and Methods

This study uses a decolonial-inspired ethnographic framework, based on deconstructing western research methods, privileging Indigenous and place-based knowledge production, and holding research accountable for its utility for underserved groups [39,40,41,42,43,44]. To better understand movement processes, data collection focused on participant observation between September 2018 to September 2020, and interviews with community members and advocates (*n* = 31) and focus groups with respondents from throughout the Central Coast (*n* =12). Community advocates played a crucial role in shaping the research questions and informing the study’s priorities and commitments. In combination with the broader project, this provides a unique insight into the contexts of Latinx immigrant and Indigenous health advocacy.

A mixed-methods ethnographic approach was used to organize multiple types of data collection. Community input was especially prioritized through participant observation, a survey of community health needs and assets, and focus groups. Participant observation included attending collective action and public advocacy efforts and developing relationships with key informants [45]. Because I designed this project to be responsive to community needs, participant observation continued during the pandemic in a primarily digital environment [42,46,47]. Members of the community hold diverse and sometimes competing views, and thus I practiced reflexivity to organize and evaluate the perspectives presented through various data collection procedures including organizational communications and online archives of group events or meetings.

My family and I have lived on California’s Central Coast for decades. I utilized aspects of my insider knowledge to gain greater insight into my observations. This helped me build relationships with individuals who do not occupy organizational leadership positions but play a key role in maintaining community relations and mobilizing grassroots actions. I shopped in locally owned stores and grocery markets, attended family celebrations, frequented public parks, participated in cultural events and holidays, accessed healthcare, and interfaced with local government agencies as a resident, constituent, and advocate. My daily investment in these communities has helped me to continuously center the need for effective and equitable solutions to healthcare disparities.

Individuals were recruited for interviews if they identified as one or both categories: immigrant community member (i.e., “Undocumented, Dreamers, mixed-status family member, resident and/or a naturalized citizen”) or advocate (i.e., “individuals that actively participate in social change efforts toward advancing immigrant health equity”). Both groups were asked questions about health needs in the community, language and cultural diversity, and perceptions of the policy environment. Participants were recruited by phone, email, flyers, and social media ads. These efforts focused on groups that provide healthcare and legal aid for immigrant communities, community-based advocacy groups, schools, and community gathering places. Snowball sampling was used to recruit participants from the networks of previous interviewees [48]. This procedure is especially useful for recruiting participants and building trust among groups that have been historically marginalized and/or exploited by government agencies. I conducted 31 interviews before reaching saturation: the scope of new information in additional interviews was exceeded by repeated information from one or more previous interviews. 

Focus groups were structured to include knowledge sharing from earlier interviews and the survey of health needs and assets. The conversations were structured around major themes in the study: political climate, health needs/assets, and language. Within each section, relevant findings were shared, and then participants had the opportunity to respond, ask questions, and discuss. Open-ended conversations set a tone for sharing knowledge and held the research accountable to the lived experiences of diverse community members [49,50]. The discussions allowed their voices to inform the interpretation of findings and set the priorities for analysis and dissemination of findings [51]. To facilitate opportunities for free-flowing discussions over Zoom, each of the six focus groups had up to four participants each. Participants were recruited from all counties represented in the study (Monterey, San Luis Obispo, Santa Barbara and Ventura). Half of the focus groups were conducted in English and the other half were conducted in Spanish (“cafecitos”) based on language of preference. Incentives were provided for each interview and focus group participant (USD 25) and key informants with both community members and advocates (USD 30) in the form of cash or a gift card.

Data were coded using a grounded theory approach [52,53] with the software, Dedoose (Dedoose Version 8.3.43, Los Angeles, CA, USA). From participant observations, data included field notes, audio memos, recorded events, public materials, organizational documents, emails, newsletters, and news media. Codes were developed by closely reading materials and capturing relevant themes in participants’ own words. The codebook developed from this stage was used for the analysis of interviews and focus groups. In continuing with the grounded theory approach, new codes were added upon comparing the interviews with field work documents [54]. A team of six research assistants were trained in coding techniques and completed line-by-line coding of each transcript. We established inter-rater reliability by meeting to discuss themes and resolve any discrepancies in the assignment of codes or creation of new codes. All items were coded by at least two research team members. When discrepancies existed, the team first sought to create consensus between the two coders. Agreement was reached in all such situations. Themes from these coding processes were then used for focused coding to interrogate social movement dynamics that bring together aspects of immigrant and health advocacy and how they changed during the pandemic.

## 3. Results

The COVID-19 pandemic has had a disparate impact on Latinx immigrant and Indigenous communities, a national trend [55,56] that is also reflected on the Central Coast [57]. Interviewees shared a range of reactions to the pandemic itself, including personal experiences and family loss. They also reflected on acts of collective resilience, which was evident in advocates’ quick response to spread information and resources to protect vulnerable community members. During the same time period, the threats to Latinx immigrant and Indigenous communities also included the proposed changes to the public charge rule and a systematic exclusion from government relief and aid programs. 

To combat this unstable environment, groups used accessible language and culturally responsive communication to promote heath literacy and share relevant COVID-19 information. Direct support efforts worked to address the legal exclusion of immigrants from federal safety net programs. The practice of mutual aid and collective responsibility enacted during the pandemic builds from traditions of solidarity and kinship within Latinx immigrant and Indigenous communities. Furthermore, advocates went beyond repairing immediate harm to preventing future risk and exploitation. Working towards changing policies influences future opportunities for mobilization and can change systems. Throughout these actions, local experts were successful in implementing change when they bridged new information with meaningful cultural frameworks shared by multiple stakeholders.

### 3.1. Multilingual Interpretation for Health Access 

Access to multilingual translation and interpretation facilitates inclusion for Spanish and Indigenous language speakers. On the Central Coast, the Mixteco/Indígena Community Organizing Project (MICOP) centralizes language interpretation in their work and provides Indigenous language access within networks of immigrant and health advocates. The group launched *Radio Indígena* in 2014, a local FM station with over 40 h of weekly live programming, featuring at least seven Mixteco languages, Zapoteco, and Purépecha. This service provides information and entertainment relevant to Indigenous farm working communities, such as support for low-income individuals to receive rental assistance, energy payment programs, and family paid leave.

This platform served a crucial role in spreading vital public health information during the COVID-19 pandemic. They quickly expanded to include Facebook Live broadcasts to supplement radio programming and demonstrate visually how testing works and show images of how others have participated in testing. These events are interpreted in at least one Indigenous language, often through consecutive interpretation. In engaging the audience, the speakers use a conversational style. The interpreter does not just repeat the words translated from Spanish but is trained to share the message in a relevant way for heterogenous audiences. For further accessibility, the videos include visual aids and photographs to guide individuals through practical steps to access transportation and where to arrive for testing. This visual demonstration helps to work against stigma and decrease fear of health services. The group emphasized overall health and wellbeing to encourage maximum prevention and safety when information about transmission was limited.

Interpretation services must be culturally responsive to regional diversity. One advocate interviewee elaborated on this, saying “Mixteco, there’s not only one variant, there’s 20 to 23, if not more. It is a big barrier, but that’s why there’s organizations like 805 UndocuFund or MICOP [Mixteco/Indígena Community Organizing Project], and IMPORTA” (Advocate Interview). MICOP created materials for World Languages Day to demonstrate regional differences between dialects from San Juan, Mixtepec, Piedra Azul, San Martín Peras, and Tlahuapa, Guerrero. Herencia Indígena is an organization that specializes in medical interpretation and partners with service providers to increase access to healthcare directly. Sharing complex medical information had increased salience during the pandemic. A *Radio Indígena* broadcast from June 2020 notified the community about a COVID-19 outbreak in H2A housing for farmworkers in Ventura County. The advocates shared information about how people could protect themselves from COVID, acknowledging the reality that social distancing is practically impossible in crowded common family and work areas. This broadcast was interpreted in Spanish, Mixteco, and Purépecha.

### 3.2. Creating Direct Support Mechanisms: Immigrants and Mutual Aid

While the virus was a central concern during the pandemic, it joined several longstanding issues affecting the community. As identified by the key informants, the 45th U.S. Presidential administration sponsored a number of hostile and xenophobic policies. This included continuing to deport and detain migrants during a pandemic, increasing construction of a Border Wall, excluding immigrants from emergency and pandemic aid programs, and seeking to expand the public charge rule. These political hostilities advance a culture of threat and fear, but advocates support an alternate vision. As MICOP put it in an informational session on the public charge rule, “¡El conocimiento es la mejor defensa contral el miedo!” (“Knowledge is the best defense against fear”). They directly helped immigrants understand how certain policies do and do not affect them. Although information sharing is vital, it cannot reverse the exclusions that do exist.

To provide a direct and clear response to these exclusions, advocates organized networks of solidarity. These were typically organized around region or county, allowing service providers and businesses to coordinate and share resources with community members. These efforts mirror how state agencies seek to create wrap-around services with multiple points of access to resources, although these groups are usually organized on a voluntary basis or subsidized by personnel from non-profit and community-based organizations. Advocates demonstrated their tenacity and resourcefulness in mobilizing under hostile conditions. One community member shared her perspective when she said, “Nuestra estrategia es más que nada seguir insistiendo, seguir tocando puertas por puertas, seguir insistiendo dándoles la información correcta. Hay muchos recursos de los que ellos se pueden beneficiar… Eso nos hace más fuertes.” (“Our strategy is more than anything to keep insisting, keep knocking door to door, keep insisting by giving them the correct information. There are many resources for their benefit… This makes us all stronger”).

The 805 UndocuFund is a resource for immigrant communities that has provided substantial benefits in the face of legal exclusions from federal programs. The program was first started in response to the 2017 Thomas Fire as a collaboration between Central Coast Alliance United for a Sustainable Economy (CAUSE), MICOP, Future Leaders of America (FLA), and Ventura County Community Foundation. They fundraised over USD 5 million for struggling workers and their families during the COVID-19 pandemic. These funds were distributed to workers in small grants of USD 1000 each, but they were not able to meet the full level of demand within the community. Similar groups such as the San Luis Obispo County UndocuSupport group distributed community-generated funding to undocumented communities throughout the Central Coast. These funds are distributed by community groups with trusting relationships in the communities, who ward against fraud and scams. 

### 3.3. Mobilizing for Farmworker Dignity 

Latinx Immigrant and Indigenous advocates are heavily involved in improving working conditions for farm working communities. For example, youth leaders from CAUSE provided a powerful testimony at the Santa Barbara and Ventura County Board of Supervisors in July of 2021. One young person spoke about her parents’ experience who were fired from the farm on the argument that there was no fruit to be harvested. She said:

We all know that’s a lie. It’s the right season to grow fruit. It’s also the right season to give the farmworkers the help they need. They are expected to work in the sun and feed this nation, how can we turn our back on them when they have been doing so much for us? How many more lives needs to be at risk before you bat an eye at them? How many more mothers are you going to take from us? I urge you to do the right thing and help them by providing masks, gloves, clean water, anything they need. They help feed our families and we should help keep their families safe. What are you going to do? Give them a helping hand, or wait until my mother or her coworkers also get COVID?

Advocates embraced and deployed the framing of farmworkers as “essential workers” to hold policymakers and farm owners accountable. The pandemic brought into sharp relief those aspects of the global economy that society cannot do without, including agriculture. Moreover, it also showed that the government can mobilize resources, funds, and services to communities in need, especially when “crises” are declared.

Policy change for farm workers includes organizational policies by individual farms, especially those that set industry standards for practices, pay rates, and conditions. For example, targets for social movement organizing included County Boards of Supervisors, as well as worker housing sites, employers, and labor contractors. During the pandemic, housing became an important site of direct advocacy to increase health regulations for testing and screening of COVID-19, as well as making opportunities available for quarantine when farm workers tested positive. Through an initiative called Housing for the Harvest, hotel rooms were made available along with services in English, Spanish, and Mixteco [58]. Because immigrants are excluded from most formal relationships with the state, employers also play a more pronounced role in influencing community members’ daily experiences.

During 2020, advocates in Santa Maria organized farm workers to obtain a pay raise and improve working conditions. The central victory at Rancho Laguna Farms included a raise of USD 2.10 per box of strawberries [59], in addition to increased access to shade for safe and socially distanced breaks as well as training for more effective communication by supervisors. After over a decade with no wage increases, this change was a historic win for organizers. Successful mobilization was a result of multiple strategies, including petitions, a work stoppage, and public demonstrations [60]. This action inspired resistance among the farm owners, who attempted to suppress organizing by contacting law enforcement officers. CAUSE was able to successfully win a settlement through the California Agricultural Labor Relations Board of USD 30,000 for 212 farmworkers in Santa Maria as compensation for damages caused by unlawful retaliation. The farm owners were additionally required to provide training on farmworkers’ rights to supervisors under the Agricultural Labor Relations Act, including the right to organize. By exercising their collective power, farm workers inspire others to honor the dignity and value created by essential workforces, during and beyond the pandemic.

## 4. Discussion

Latinx Immigrant and Indigenous Health movements demonstrated adaptability in the pandemic environment by building from existing resources and networks of solidarity. These strategies were culturally responsive and language accessible and provided support resources to the community. These strategies are characteristic of both health access movements and constituency-based movements [5,6,61,62]. However, the efficacy of information is limited by structural constraints such as housing conditions, racism, and xenophobic policies [63,64,65]. Advocacy and direct action disrupted patterns of exploitation and mobilized supportive decision makers at the local and state levels [30,31,32,37,66,67]. The state of California polarized against the federal government’s hostilities toward immigrants, creating political openings for policy change. For instance, advocates won Medi-Cal expansion to include income-eligible minors and young adults up to age 26.

This study’s strengths are rooted in the decolonial-inspired framework, including the use of multiple forms of data, the level of rapport between the researcher and the community, and the deployment of locally-specific explanations of events and perspectives. However, there are also some study limitations, especially in terms of time and scope. Future researchers would benefit from examining these dynamics over time in other communities of Latinx immigrant and Indigenous people. While some basic aspects of temporal change were observed between 2018 to 2020, this study is limited to the COVID-19 pandemic and its early days in the United States. More research is needed to understand how these strategies evolved over time. 

Immigrant health movements depended on grassroots mobilization to build trusting relationships between community leaders and those most in need. Due to historic systems of exploitation, Latinx communities mistrust government agencies and mainstream healthcare providers. The shared experiences of structural exclusion shape an embodied experience of health risk [68], so that being an “essential worker” becomes politicized as a condition of health risk. This demonstrates how health social movements mobilize collective identities beyond illness itself [5,27]. This mobilization was effective because of the expertise of community leaders who advocate for systems change at every level.

Legislative change, organizational policy, and local actions can further improve the health of Latinx Immigrant and Indigenous communities. These recommendations build from the movement pandemic adaptability of local organizers and state-level advocates. In California, state legislation can lead the way for national change that is more inclusive and equitable for Latinx Immigrant and Indigenous communities. State legislators have recently expanded state public health insurance (Medi-Cal) to undocumented elders aged 50 and up. Policymakers are poised to further expand Medi-Cal to include undocumented adults of all ages, with the Governor’s draft budget for 2022–2023 funding coverage to start in 2024. If this moves forward as planned, California will be the first state to remove immigration status as a barrier to public health insurance. However, insurance and access only addresses part of the issue.

City, county, state, and federal governments can declare racism a public health crisis and take immediate action accordingly. Taking aggressive action against racism means immediately ending practices that reproduce health disparities. For instance, policymakers can disallow county sheriffs and city police from coordinating with federal immigration agencies such as Immigration and Customs Enforcement. Appropriate action also includes providing funding, resources, and capacity support for anti-racist and immigrant-serving organizations. Local groups can contribute to equity and access by regularly hiring multilingual interpreters. In a medical context, this action is necessary to ensure access to care for all individuals. In all contexts, translation and interpretation services need to be responsive to local audiences, including Indigenous language speakers.

## 5. Conclusions

Researchers have documented the widespread existence of health inequities for Latinx immigrant and Indigenous communities. Discrimination and systemic oppression are major drivers behind these realities. Scholars have also examined the successful advocacy strategies of immigrant rights groups, as well as constituencies invested in improving healthcare access and outcomes. This paper brought together research on these topics to examine the specific dynamics of immigrant health movements in the context of the COVID-19 pandemic. The concept of “movement pandemic adaptability” highlights the implementation of culturally responsive strategies for healthcare rooted in collaborations between diverse community leaders and healthcare experts. Future research is needed to extend this concept to examine other community pandemic responses, as well as to evaluate the role of responsive community-based strategies within public health systems in a range of crisis settings. 

## Data Availability

Reports of the data presented in this study are available in English and Spanish online at https://lagenteunidacc.wixsite.com/recursosdesalud/reports-reportes, accessed on 21 July 2022.

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
