# Peer review of "Movement Pandemic Adaptability: Health Inequity and Advocacy among Latinx Immigrant and Indigenous Peoples"

_ijerph, 2022, doi:10.3390/ijerph19158981_

Round 1

Reviewer 1 Report

Thank you for the opportunity to review this paper. This manuscript presents a robust ethnography study that can be highly informative not just for policies regarding Latinx Immigrant Health and Indigenous Health but also for other ethnography studies with a similar scope. My major concerns lay within the introduction and discussion. The introduction is too lengthy and detailed while not providing a thorough review of other research on the topic. While the ideas are well-presented, the level of detail makes it difficult to discern the study aim. The discussion does not dive into the significance of results in the context of other studies and current public health policies. With changes to the introduction and discussion, this manuscript may be much improved. Please see detailed comments below.

Introduction

While this introduction is very detailed and explains the current policy landscape and the author’s conceptual work, there is not a thorough review of the literature on research already completed on this topic or similar topics. If no other literature exists, this should be stated clearly. Furthermore, the introduction is quite long. Much of the information is repetitive and can be made more concise. The length of the introduction may convolute the research aim for readers.

Line 36: “chapter” should be changed to “paper”

Line 52: Another reference to a chapter

Line 72: While the need to end systemic racism can be deduced from a reader’s broader knowledge of health care outcomes, the case for focusing research on ending systemic racism is not clearly laid out. It would be helpful to link the potential health outcomes that could be improved with research focusing on ending systemic racism. It may also be helpful to discuss drawbacks of the current research documenting inequities.

Line 74: Another reference to “chapter”

Line 135: “chapter” reference

Methods

Methods overall are robust and are a very interesting approach to address the research question. I especially appreciated the inclusion of materials beyond interview transcripts and observations—e.g., public materials, documents, emails, newsletters, and news media.

In general, tracking one’s positionality throughout the research is useful in qualitative research.  For ethnography and other qualitative publications, it is common to include a summary of the researcher’s positionality in a short statement. It can help the reader understand and discern any biases that could affect results. I suggest adding this is in.

Line 149: “chapter” should be replaced

Line 175: “chapter” should be replaced

Lines 197-200: Please elaborate on methods to ensure inter-rater reliability and the coding process. For one, how many coders coded each set of materials? 2? All 6? In addition, when meeting to discuss discrepancies, how were discrepancies resolved? Did all 6 coders need to agree? Or did only 1 additional coder needed to weigh in?

Results

Results are thoroughly summarized. It would be helpful to include a couple more quotes from the data that can support the summaries.

It would be a helpful addition to include some data visualization. Perhaps a word cloud? As it stands, the manuscript is quite text heavy.

Lines 262-263: While it may be a point of view of the author, political commentary from the author’s own point of view is typically less appropriate in research publications. It can indicate researcher bias in result interpretation. However, it is possible that this conclusion arose from the data and not from the author’s opinion. If this is true, it should be stated as such.

Discussion

The discussion provides conclusions supported by the data. However, it could be developed more. Results should be placed in the context of specific policy changes and the current public health landscape. How do the results compare to the results of other similar studies?

Line 354: “Outright hostilities” is a conclusion that does not seem to arise from the data but instead express the author’s opinion.

Conclusions

The presented conclusions seem to be more appropriate for the discussion section. The conclusions should be a concise statement of the takeaways from the study as opposed to discussions of specific policies.

Reviewer 2 Report

Title :

Needs to be improved ( should adequately describe the contents and / or purpose of your study
It should contain important  key words , such as inequalities in health care .

Introduction:

It is too long ( economy of words is important in scientific literature )
State the problem, why it needs to be addressed , what work has already been done on the subject
And what are gaps
Acknowledge the research work of others on the subject
Finally describe the objectives of your study

Methods:

The interviews were conducted with 31 Key informants ( community members and advocates .
What was selection criteria ?  Why 31 key informants ?
Who conducted the interviews ?   

The participants were observed and findings were documented . Was there any check list for observations ?

Focus groups :

How many FGDs were conducted ?
Why each FGD comprised of only four participants?
How participants for FGDs were selected ?
How FGDs were conducted? ( means team members who conducted the FGDs)

Discussion:

too short.
Can be improved
Summaries and discuss your main findings
Give your interpretation and discuss the implications

Limitations of study :
Acknowledge the limitations of your study

Round 2

Reviewer 1 Report

I appreciate the responsiveness to the comments. I am satisfied with the responses to my feedback and the edits made to the manuscript. In particular, the introduction provides excellent background and describes the significance of this work.

Author Response

Thank you for your review of the paper! I appreciate the feedback that made this a stronger manuscript.